# FeatureOnto: A Schema on Textual Features for Social Data Analysis

Sumit Dalal[1*], Sarika Jain[1] and Mayank Dave[2]

[1] Department of Computer Applications, National Institute of Technology, Kurukshetra, India
[2] Computer Engineering, National Institute of Technology, Kurukshetra, India
sumitdalal9050@gmail.com

**Abstract.** Social media is one of the valuable information sources which present much data to the researchers. This information is mainly analyzed by machine learning and the deep learning methods, which lack semantics and interpretation in their outputs. Also, much attention is paid to the feature engineering there. We present a taxonomy of the different feature categories. The categories relate to the features learned during the training for analyzing the textual information, specifically available on social platforms. The ontological view of the data will represent knowledge in a more understandable form besides interpreting the machine learning results for various tasks related to the social data analysis. We chose Depression as the use case purpose. The ontology is designed using Ontology Web Language and Resource Description Framework in the Protégé. The validation of the ontology is carried out with designed competency questions.

**Keywords:** Deep Learning, Depression, Knowledge Graph, Machine learning, Ontology, Social Data, Twitter.

## 1    Introduction

Mental health is an essential aspect of human life to live a productive and energetic life. People tend to ignore their mental health for various reasons like inaccessible health services, limited time for themselves, etc. Nevertheless, technological advancements provide researchers an opportunity for pervasive monitoring to include users' social data for their mental health assessment without interfering with their daily life. People share their feelings, emotion, daily activities related to work and family on social media platforms (Facebook, Twitter, Reddit). These posts can be used for extracting features or looking for particular words, phrases that can be used to assess if a user has depression or not.

Various machine learning and deep learning methods have been devised and applied for mental health assessment from users' social data. These techniques mainly consider correlation or structural information of the text for classification purposes. They miss contextual information of the domain. Analyzing social data with traditional statistical and machine learning approaches has limitations like poor big data handling capacity, semantics, and contextual/ background knowledge inclusion. Recently deep learning approaches have been widespread, but interpretability is a significant issue.

So a hybrid approach that handles semantics and big data should be considered for better results.

Contextual information can be represented by a logic-based model [McCarthy, J. 1993], Key-Value pair [Schilit, B., 1993 ], object-oriented model [Schmidt, A., 1999], UML diagram [Sheng, Q. Z., & Benatallah, B. 2005], or markup schema. Nevertheless, these models have limited capacity in representing real-world situations. We propose to develop an ontology to represent the domain information. Ontology is a formalization of a domain's knowledge [Gruber, T. R. 1995]. The main principles of ontology are to reuse sharing domain knowledge between agents in a language understandable to them (user or software). Ontology has been developed and used in different application domains. [Konjengbam A.2018, Wang D. 2018] design ontology for analyzing user reviews in social media. [Malik S. & Jain S. 2021 and Allahyari M. 2014] employ ontology for text documents classification while [Taghva, K., 2003] uses ontology for email classification. [Dutta B. & DeBellis M. 2020, Patel A. 2021] developed an ontology for collecting and analyzing the covid-19 data. [Magumba, M. A., & Nabende, P. 2016] develop an ontology for disease event detection from Twitter. [Chowdhury, S., & Zhu, J. 2019] use topic modeling methods to extract essential topics from transportation planning documents for constructing intelligent transportation infrastructure planning ontology. However, ontology-based techniques for depression classification and monitoring from social data have been insufficiently studied.

The machine learning and statistical approaches consider limited contextual information. Moreover, it is not easy to interpret their results. For this reason, deep learning models are considered complete black boxes. Personalization of the system is another issue that needs to be in focus. For the implementation purpose, we chose depression as the domain. We aim to develop an underlying ontology for the personalized and disease-specific knowledge graph, to monitor a depressive user through his publicly available textual social data. The ontology is designed using Ontology Web Language and Resource Description Framework in the Protégé. The validation of the ontology is carried out with designed competency questions.

Our Contributions in this paper are as follows:

1.      To develop the FeatureOnto ontology for analyzing social media posts. The features of social posts manipulated by machine learning and deep learning techniques are arranged in a taxonomy. This way, structured data help in the interpretation of output produced.

2.      We write competency questions to describe the scope of FeatureOnto to detect and monitor depression through social media posts.

The remaining paper is organized into four sections. Section 2 discusses the related literature. The FeatureOnto development approach and its scope will be discussed in section 3. Section 4 discusses the conceptual design of the FeatureOnto and the evaluation of the same. Conclusion and future work is discussed in the last section.

## 2    Literature

This section discusses previous research on depression ontology development using various sources or employing the available ontology for depression detection or monitoring.

### a.    *Ontology Based Sentiment Analysis.*

Sentiment analysis is a crucial aspect in detecting depression from social posts. However, there are applications other than mental health assessment where it is functional. Sentiment extraction of user posts/reviews is a popular application that considers the affective features of the posts. The authors consider eight emotion categories to develop an emotion ontology for the sentiment classification [Sykora, M., 2013]. [Saif, H., 2012] employ entity extraction tools for extracting entities and mapping semantic concepts from user reviews. They use the extracted semantic features with unigrams for Twitter sentiment analysis. [Kardinata E. A. 2021 and] apply the ontology-based approach for sentiment analysis.

### b.    *Ontology in Healthcare Domain.*

In the healthcare domain, ontologies have been employed for quite a long time. [Batbaatar, E., & Ryu, K. H. 2019] employ Unified Medical Language System (UMLS) ontology to extract health-related named entities from user tweets. [Krishnamurthy, M. 2016] implement DBpedia, Freebase, and YAGO2 ontologies for determining behavior addiction category of social users'. In [Kim, J., & Chung, K. Y. 2014], authors develop ontology as a bridge between the device and space-specific ontologies for ubiquitous and personalized healthcare service environment. [Lokala, U.,2020] build ontology as a catalog of drug abuse, use, and addiction concepts for the social data investigation. [On, J., 2019] extract concepts and their relations from clinical practice guidelines, literature, and social posts to build an ontology for social media sentiment analysis on childhood vaccination. [Alamsyah, A., 2018] build ontology with personality traits and their facets as classes and sub-classes, respectively, for personality measurements from Twitter posts. [Ali, F., 2021] design a monitoring framework for diabetes and blood pressure patients that consider various available ontologies medical domain and patient's medical records, wearable sensor and social data.

### c.    *Depression monitoring & Ontology.*

Authors employ ontologies in depression diagnosis and monitoring. Either they build ontology or use available ones. [Martín-Rodilla, Patricia 2020] propose to add temporal dimension in ontology for analyzing depressed user's linguistic patterns and ontology evolution over time in his social data. [Benfares, C. 2018] represents explicitly defined patient data, self-questionnaire and diagnosis result in the semantic network for preventing and detecting depression among cancer patients. [Birjali, M.

2017] constructs a vocabulary of suicide-related themes and divides them into sub-classes concerning the degree of threat. WordNet is further used for semantic analysis of machine learning predictions for suicide sentiments on Twitter. Some works that build an ontology for depression diagnosis are discussed below. We assign ontologies unique ids such as O1, O2, etc. These ids are used in table 2 to mention the particular ontology.

**O1.** [Petry, M. M. 2020] provides a ubiquitous framework based on ontology to assist the treatment of people suffering from depression. The ontology consists of concepts related to the user's depression, person, activity, and depression symptoms. Activity has subclasses related to the social network, email, and geographical activities. The person has a subclass PersonType which further defines a person into User, Medical, and Auxiliary. We are not sure if a patient history is considered or not.

**O2.** [Kim, H. H., 2018] extract concepts and their relationships from posts on the dailyStrength, to develop the OntoDepression ontology for depression detection. They use tweets of family caregivers of Alzheimer's. OntoDepression has four main classes: Symptoms, Treatments, Feelings, and Life. The symptom is categorized into general, medical, physical, and mental. Feelings represent positive and negative aspects. Life class captures what the family caregivers' are talking about. Treatments represent concepts of medical treatment.

**O3.** [Jung, H., 2016/2017] develop ontology from clinical practice guidelines and related literature to detect depression in adolescents from their social data. The ontology consists of five main classes: measurement, diagnostic result & management care, risk factors, and sign & symptoms.

**O4.** [Chang, Y. S., 2013] build an ontology for depression diagnosis using Bayesian networks. The ontology consists of three main classes: Patient, Disease, and Depression_Symptom. Depression symptoms are categorized into 36 symptoms.

**O5.** [Hu, B., 2010] developed ontology based on Cognitive Behavioral Theory (CBT) to diagnose depression among online users at the current stage. Their focus is to lower the threshold access of online CBT. The ontology consists of the patient, doctor, patient record, and treatment diary concepts.

Work in [Cao, L., et. al. 2020] created ontology for social media users to detect suicidal ideation from their knowledge graph. Their work is similar to our work but they considered limited features taxonomy, moreover we focus depression detection from personalized knowledge graph.

Table 2 compares distinct ontologies built in different research papers to detect and monitor depression are compared on four parameters (Main Classes, Dimensions Covered, Entities Source Considered, and Availability & Re-usability). We extracted seven dimensions (Activity, Clinical Record, Patient Profile, Physician Profile, Sensor Data, Social Posts, and Social Profile) from the related literature. A description of each dimension, along with dimension ID, is given in table 1. We cannot find the ontologies built by other authors online. We are not sure if these are available for re-

use or not, so the Availability & Re-usability column is blank. O1 ontology has scope over almost all the dimensions we have considered.

**Table1.** Description of different dimensions considered

| Dimension | Dimension ID | Description |
|---|---|---|
| Activity | D1 | This facet covers physical movements, social platforms, daily life activities etc. |
| Clinical Record | D2 | It is related to patient profile, provides historical context, and covers clinical tests, physician observations, treatment diary, schedules, etc. |
| Patient Profile | D3 | The dimensions cover disease symptoms, education, work condition, economical, relationship status, family background etc. |
| Physician Profile | D4 | This aspect describes a physician in terms of his expertise, experience, etc. |
| Sensor Data | D5 | This element is related to the smartphone, body, and back-ground sensors. |
| Social Posts | D6 | It is affiliated with the content of posts by a user on SNS. |
| Social Profile | D7 | Social media profile provides an essential aspect of user personality. |

**Table2.** Comparison of the FeatureOnto and depression ontologies used in literature

|  | Main Classes | Dimensions Covered | Entities Source Considered | Availability & Re-usability |
|---|---|---|---|---|
| O1 | Depression, Symptom, Activity | D1, D2, D3, D4, D5, D6 | Literature | --- |
| O2 | Symptoms, Treatments, Life, Feelings | D1, D6 | SNSs | --- |
| O3 | Diagnostics, Subtypes, Risk Factors, Sign& Symptoms, Intervention | D6 | CPG, Literature, SNSs, FAQs | --- |
| O4 | Patient, Disease, Symptom | D2, D3 | Literature | --- |
| O5 | Patient, Doctor, Activity, Diagnosis, Treatment Diary | D1, D2, D3, D4 | General Scenario | --- |
| Our Approach | Patient, Symptom, Posts, User Profile, Feature | D2, D3, D6, D7 | Literature | Yes |

## 3  Designing FeatureOnto Ontology

The focus of ontology development is to analyze the social textual data and interpret the results produced by the machine learning or deep learning models. Mainly, authors focus on n-gram features of social media posts, but FeatureOnto also considers other features. We follow the 'Ontology Development 101' methodology for FeatureOnto development [Noy, N. F., & McGuinness, D. L. 2001]. An iterative process is followed while designing the ontology lifecycle.

**Step 1.  Determining Domain and Scope of the Ontology**
We create a list of competency questions to determine the ontology's domain and scope [Grüninger, M., & Fox, M. S. 1995]. FeatureOnto ontology should be able to answer these questions. E.g., What are the textual features of social media posts? The ontology will be evaluated with these questions. Table 3a and 3b provide the sample of competency questions where 3a is derived to check the ontology schema, i.e., ontology without any instance. In comparison, questions in 3b are derived keeping in mind the use case of depression monitoring of a social user. Queries of table 3b are out of scope for this paper as here we are only presenting the schema.

**Table3a.** Schema Based Competency Questions.

| Competency Questions |
| --- |
| 1. Retrieve the labels for every subclass of the class Content? |
| 2. "Topics" is the subclass of? |
| 3. What type of feature is "Anger"? |

**Table3b.** Knowledge Graph Based Competency Questions.

| Competency Questions |
| --- |
| 1.  What is the sleeping pattern of a user/patient (user can be normal patient)? |
| 2.  In which hour user messages frequently? |
| 3.  How many posts has low valence in a week? |
| 4.  Emotional behavior pattern considering week as a unit? |
| 5.  Daily/weekly average frequency of negative emotions? |
| 6.  Compare daily/weekly/overall average number of first person pronoun and second/third person pronouns? |
| 7. What are the topics of interest for a depressed user? |
| 8. Anger related words used frequently or not? |
| 9. Find the pattern of psycholinguistic features? |

10. What symptoms a user is having?
11. Compare posting pattern of depressed and non-depressed social user?

**Step 2. Re-using the Existing Ontologies**

We search for available conceptual frameworks and ontologies on social data analysis at BioPortal [Musen, M. A. 2012], OBOFoundary, and LOD cloud. Ontologies representing sentiment analysis or depression classification, or other social media analysis tasks on the web (Google Scholar, Pubmed) and the kinds of literature are searched for the required concepts and relationships. We have done a comprehensive search but could not find a suitable ontology that could be re-used fully. We find some ontology and can inherit one or more classes from them. Most of the inherited classes are given attributes as per our requirements. Table 4 shows our efforts toward implementing the reusability principle of the semantic web. Figure 2, present in the next section, gives a diagrammatical representation of the inherited entities. Different colors define each schema. The solid and the dotted line show immediate and remote child-parent relations between classes. Most inherited entities belong to Schema, MFOEM, and HORD, while APAONTO, Obo are the least inherited ontologies. We did not find suitable classes for UniGrams, BiGrams, Emoticon, and POSTags. So we use our schema to represent these classes. The solid and the dotted line represent the property and the subclass relationship between two entities.

**Table4.** Entities and Namespaces considered in the FeatureOnto.

| Entity | Sub Entities | Schema Selected | Available Schemas |
|---|---|---|---|
| Content | UniGrams, BiGrams, POSTags | --- | --- |
| Emoticon | --- | --- | --- |
| Emotion | Arousal, Positive, Negative | MFOEM | MFOEM, SIO, VEO. |
| | Dominance | APAONTO | APAONTO, FB-CV. |
| GenderType | --- | Schema | Schema, GND |
| Person | Patient | Schema | FOAF, Schema, Wikidata, DUL. |
| | User | HORD | NCIT`, SIO, HORD. |
| Post | --- | HORD | HORD |
| Psycholinguistic | Anger, Anxiety, Sad | MFOEM | MFOEM, SIO, VEO, NCIT. |
| | Pronoun | --- | --- |
| Symptoms | --- | Obo | NCIT, SYMP, RADLEX, Obo |
| Topic | --- | | EDAM, ITO |

**Step 3. Extracting Terms and Concepts**

Keeping in mind our use case, we read literature on depression and mental disorders detection from social data using machine learning or lexicon-based approaches and extract terms related to features considered for classification. We found that different textual features are extracted and learned in machine learning or deep learning training phase [Dalal, S., 2019, Dalal, S., & Jain, S. 2021], e.g., bigrams, unigrams, positive or negative sentiment words. Table 4 shows different entities and sub-entities present in the FeatureOnto ontology. It also provides information about the various available schemas for an entity and the schema used for the inheritance. We also search social networking data to extract additional terms. The extracted terms are used for describing the class concepts.

### Step 4. Developing the Ontology and Terminology

We have defined the classes and the class hierarchy using the top-down approach. The ontology is developed using Protégé [Musen, M. A. 2015]. The ontology is uploaded on BioPortal.

### Step 5. Evaluating the Scope of the Ontology

A set of competency questions is given in Tables 3a and 3b. For scope evaluation of the FeatureOnto, answers to the SPARQL queries built on the questions from table 3a are considered. Results of the queries are discussed in the coming sections.

## 4    FeatureOnto Ontology Model

Following the steps discussed in the previous section, we design FeatureOnto ontology. A high-level view of the FeatureOnto ontology is represented in Figure no. 1 Complete FeatureOnto structure (at the current stage) has five dimensions (Patient, Symptom, Posts, User Profile, and Feature) covered by various classes in the figure. Most of the entities in our ontology belong to the Social Post dimension. The solid and the dotted line represent the property and the subclass relationship between two entities. Figure 1 gives a conceptual schema of the proposed model. FeatureOnto uses existing ontologies to pursue the basic principle of ontology implementation. Figure 2 represents the terms inherited by FeatureOnto from available schemas. Different colors define each schema. The solid and the dotted line show immediate and remote child-parent relations between classes. Most inherited entities belong to Schema, and MFOEM, while FOAF, and APAONTO are the least inherited ontologies.

### Scope Evaluation of the FeatureOnto.

Table 3a and 3b presents the competency questions related to the schema and instances. This work is related to the building of the schema only, and hence we executed queries on schema only. Below, queries are built on questions from table 3a.

*Question1.* Retrieve the labels for every subclass of sf:Content*?*

*Query.* PREFIX rdfs: <http://www.w3.org/2000/01/rdf-schema#>
PREFIX sf: <http://www.domain.com/your/namespace/>Results.

SELECT ?subClass ?label WHERE {
    ?subClass rdfs:subClassOf sf:Content .
    ?subClass rdfs:label ?label .
}

*Results.* **POSTags, UniGrams, BiGrams**

*Question2.* "Topics" is the subclass of (find immediate parent)?

*Query.* PREFIX rdfs: <http://www.w3.org/2000/01/rdf-schema#>
PREFIX ns: <http://www.domain.com/your/namespace/>

SELECT ?superClass WHERE { ns:Topics rdfs:subClassOf ?superClass .
}

*Results.* **Feature.**

*Question3.* What type of feature is "Anger" (find all parents)?

*Query.* PREFIX rdfs: <http://www.w3.org/2000/01/rdf-schema#>
PREFIX ns: <http://www.domain.com/your/namespace/>

SELECT ?superClass WHERE { ns:Topics rdfs:subClassOf* ?superClass
. }

*Results.* **Psycholinguistic, Feature.**

Ontology is still under construction, when but prototype is available on https://github.com/sumitnitkkr. For generalization we have not mentioned any namespace here for our own entities.

**Figure1.** Conceptual Design of the FeatureOnto.

**Figure2.** Classes Inherited from the available Ontologies.

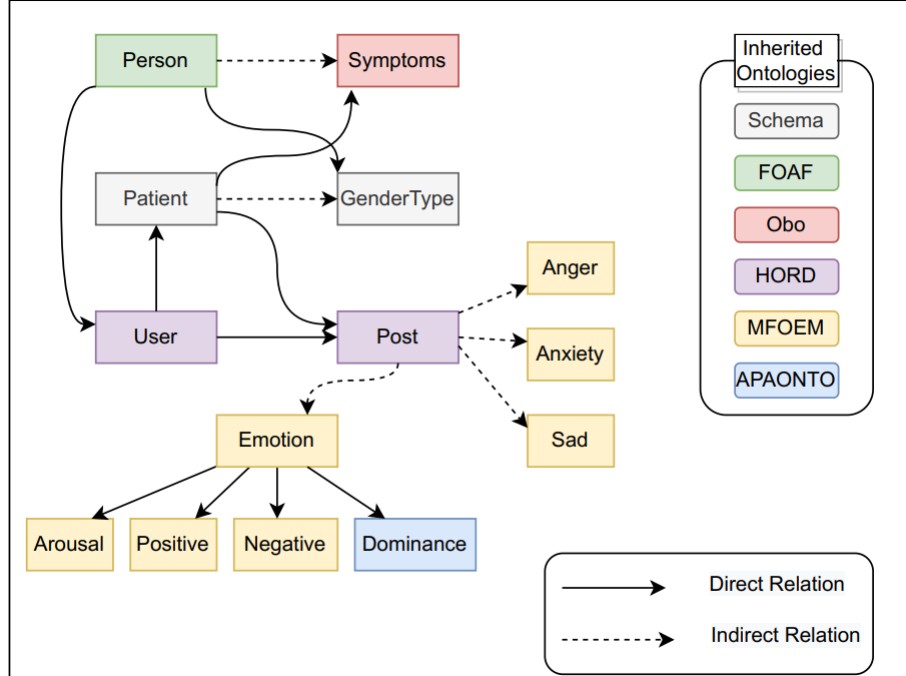

## CONCLUSION

We developed the FeatureOnto ontology to provide a taxonomy of social media posts' features (use mental health assessment or depression classification/ monitoring is taken). Posts carry huge information regarding many aspects. This information can be placed into different feature categories. These features are widely used in sentiment analysis, mental health assessment, event detection, user profiling, document classification, and other natural language and image processing tasks. The ontology will be used to create a personalized depression knowledge graph in the future. For this reason, it does not focus on the concepts from clinical practice guidelines and depression literature at the current stage. We will also extend the ontology to include other concepts related to depression in the future.

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
