# OpenReview forum: "FeatureOnto: A Schema on Textual Features for Social Data Analysis"
_kg-construct.github.io/KGCW/2022/Workshop — Submitted to KGCW 2022_

### Official Review · ~Maxime_Lefrançois1 · 2022-03-23
**not in the scope**

**Rating:** 3
**Confidence:** 5

**Review:**

This paper describes an ontology developed following the Ontology Development 101 methodology, implemented by hand on Protégé, evaluated  with a few competency questions and three very generic SPARQL queries.
The work is well positioned wrt other ontologies in the domain,
A link to a github account is given, but the ontology is not there, and not yet published following the best practices.

Most importantly, the paper is not in the scope of the KGC workshop.
KGC is about automated construction of knowledge graphs. Not manual development of ontologies.
The paper only ends with: "the ontology will be used too create a knowledge graph in the future"

---

### Official Review · ~Ana_Iglesias-Molina2 · 2022-03-27
**Premature and unfinished ontology**

**Rating:** 3
**Confidence:** 5

**Review:**

This paper describes FeatureOnto, an ontology for representing features of social media posts, and with the purpose of helping interpret the output of deep learning models that analyse these posts. It has been developed following the guidelines provided by the Ontology Development 101 methodology, implemented in Protege and evaluated with SPARQL queries.

Even though the motivation of the paper is interesting, the ontology is neither finished (as stated by the authors) nor available (contrary to what it said in the text). The text states that the ontology has been uploaded to BioPortal and is available on GitHub, and I couldn’t find it in any of the platforms. Moreover, the SPARQL queries are scarce and too simple, and don’t cover the entire scope of the ontology, thus the evaluation is insufficient. The fact that the ontology has no URI and is not published online and available hinders the reproducibility of the results provided, sends the clear message that it is not finished. I recommend the authors to keep on working on it and to take a look at the LOT methodology [1], that provides more current guidelines to create an ontology, also to publish and maintain it.

Moreover, the scope of the paper doesn’t seem completely accurate. Authors repeat throughout the paper that they are focused on the depression use case, but this is not reflected on the ontology. Ontologies can specify in the requirements specifications use cases and user stories; depression is a legit one, but the ontology presented is more general than it. Focusing the paper on the depression use case when there is in fact no relation to it in the ontology or SPARQL queries or examples of use makes it confusing without a proper argumentation.

As a conclusion, I consider that the work is interesting and promising but premature. It lacks the completion of the ontology, a more complete evaluation, and its publication to make the ontology available. Furthermore, adding examples and specific details of the use cases, and an analysis of how this can improve social media analysis in combination with the deep learning methods that authors point out would greatly improve the work.


[1] Poveda-Villalón, M., Fernández-Izquierdo, A., Fernández-López, M., & García-Castro, R. (2022). LOT: An industrial oriented ontology engineering framework. Engineering Applications of Artificial Intelligence, 111, 104755.

---

### Official Review · ~Ben_De_Meester1 · 2022-03-29
**Early-stage work with too little proof. Once improved, it would fit better in other venues**

**Rating:** 3
**Confidence:** 5

**Review:**

This seems to be a very early-stage work that proposes a new ontology for describing (textual) features in social media posts,
future work is to futher refine this for depression and related domains.
No evidence is provided to showcase the usefullness of this ontology (it's not clarified where the competency questions come from),
nor is there any proof of the technical validity (no version was provided to the reviewers nor available online).
In all, it's hard for mee to see the novelty of this work overall, also, the link with the workshop call text is not clear.
There are quite some workshops at ESWC 2022 that cover NLP, Biomedical, or ontologies,
and it's currently not clear for me why KGCW was targeted, focussed on KG Construction (from existing data), not schemas/ontologies.
Below some more detailed comments.

### Introduction

I'm a bit doubtful to read "However, ontology-based techniques for depression classification and monitoring from social data have been insufficiently studied." if you can state 8-ish ontologies in your SOTA section.

### SOTA

SOTA seems quite extensive (especially for a workshop paper), but I missed the takeaway messages: which SOTA is most relevant and why, how did this influence the design choices made in the paper?

### FeatureOnto Design

It's a pity that no reason was given to why the past 20 years of ontology engineering were not taken into account, e.g. [eXtreme Design](http://extremedesign.info/), [MODL](https://daselab.cs.ksu.edu/content/modl-modular-ontology-design-library), [LOT](https://lot.linkeddata.es/).

How was the Domain and Scope determined? Expert group, single researcher, ....?
I'm not really convinced about the quality of the competency questions: these are very much technology-oriented (terms such as 'subclass'), and not domain-oriented.
I actually assume that Table 3b is the most important table to develop the ontology from, these in fact _are_ domain-oriented.
(Even though, some questions in Table 3b are a bit too vague for me, e.g. "Compare posting pattern of depressed and non-depressed social user" -> compare how? time, frequenc, burst or not, etc...)

Considering re-use, a quick search on [lov](https://lov.linkeddata.es/dataset/lov/terms?q=part-of-speech) gives some options for, e.g. POSTag.
A reference to a more detailed explanation on why certain design choices were made would be relevant here.

### FeatureOnto Model

Given my reservations on Table 3a, I'm not convinced the presented evaluation is of any value.
Providing some dummy data and evaluating Table 3b would give more evidence of the use of the proposed schema.
Also the lack of any technical results (no published ontology findable online) makes me think that this submission is too early-stage:
I don't really consider proposing a new ontology as novelty, except when it's in use, or novel ontology engineering techniques or methods or tooling are presented,
e.g. to decrease the generation/documentation/publication effort.

---

### Decision · Program_Chairs · 2022-04-11

**Decision:**

Reject

**Comment:**

Dear authors,

Thank you for submitting your paper. Unfortunately we don’t accept your paper now in its current state. We refer to the reviews for suggestions on how you can improve your paper.

Kind regards
Organizers of the Knowledge Graph Construction workshop 2022